# Decompose Auto-Transformer Time Series Anomaly Detection for Network Management †

**Bo Wu** [1,2,3], **Chao Fang** [3,4,*], **Zhenjie Yao** [3,5], **Yanhui Tu** [3,6] and **Yixin Chen** [3,7]

1  School of Software, Nanchang Hangkong University, Nanchang 330063, China
2  Key Laboratory of Computer Network and Information Integration, Ministry of Education, Southeast University, Nanjing 211102, China
3  Purple Mountain Laboratories, Nanjing 211111, China
4  School of Information and Communication Engineering, Beijing University of Technology, Beijing 100124, China
5  Institute of Microelectronics, Chinese Academy of Sciences, Beijing 100029, China
6  Shandong Future Network Research Institute, Jinan 250002, China
7  Department of Computer Science and Engineering, Washington University in St. Louis, St. Louis, MO 63130, USA
*  Correspondence: fangchao@bjut.edu.cn
†  Opening Foundation of Key Laboratory of Computer Network and Information Integration, Ministry of Education, Southeast University (K93-9-2021-05).

**Abstract:** Time series anomaly detection through unsupervised methods has been an active research area in recent years due to its enormous potential for networks management. The representation and reconstruction of time series have made extraordinary progress in existing works. However, time series is known to be complex in terms of their temporal dependency and stochasticity, which makes anomaly detection difficult. To this end, we propose a novel approach based on a decomposition auto-transformer networks(DATN) for time series anomaly detection. The time series is decomposed into seasonal and trend components, and renovated as a basic inner block deep model. With this design, transformers can decompose complex time series in a progressive manner. We also design an auto-transfomer block that determines dependencies and representation aggregation at the sub-series level based on series seasonal and trend components. Moreover, the complex transformer decoder is replaced by a simple linear decoder, which makes the model more efficient. Extensive experiments on various public benchmarks demonstrate that our method has achieved state-of-the-art performance.

**Keywords:** networks management; time series; anomaly detection; series decompose; transformer

## 1. Introduction

Time series anomaly detection, which is a widely used technology [1,2], plays an increasingly important role in academia and industry, such as network management. Anomaly detection refers to identifying rare items, events, or observations in data that are significantly different from expectation. Network management focuses on the optimal operation of resources to ensure the rational use of performance indicators and resources. Anomaly detection is an important part of network management. the effectiveness and function of anomaly detection is directly related to the effectiveness of network management. In order to improve the availability and reliability of the network, when the network is abnormal, the network manager can find the anomaly as soon as possible and take measures in advance to avoid the impact on the service. Therefore, efficient anomaly detection technology can improve the network intelligent operation.

Time series anomaly detection has become increasingly difficult in practice due to the explosion of raw data that can be captured. In recent decades, numerous efforts have been made to detect time series anomalies. Statistical methods, supervised learning methods, and unsupervised learning methods can be roughly divided into three categories. In the

first class of methods, systems produce normal data based on a family of models, and the model parameters are learned using the normal data.

A time series anomaly detection problem is typically treated as a classification problem in supervised learning methods. Through shallow or deep models, they extract various features and perform two-class or multiclass classification. Although the above two kinds of methods provide impressive and promising results, unsupervised time-series anomaly detection methods have drawn much attention in recent years due to two main reasons. Despite the availability of large numbers of time series, obtaining the corresponding label information is difficult. It is a time-consuming and expensive process to manually annotate labels by domain experts. The anomalies are sparse even with labels, and they cannot cover every type of anomaly. Both normal and anomalous samples are highly imbalanced, which makes supervised methods inefficient. The core idea of unsupervised detection methods is to learn the normal pattern of data using a large amount of available normal data.

There may be a few serious abnormal samples in the training data, which adversely affect the learning of normal patterns, although many of the training data are normal samples. Therefore, unsupervised learning methods are usually employed to handle this task due to anomaly diversities, the lack of positive samples, and the high cost of annotations.

In practice, time-series anomaly detection has become increasingly challenging because of the explosion of raw data that can be captured. The solution to this problem has been addressed extensively over the past several years. In the early stage, anomalies were detected using classical machine learning techniques such as Gaussian models. Handcrafted features at the low level, however, cannot represent data in high dimensions. As deep neural networks have developed rapidly over the past few years, they have made significant progress. It is now possible to learn highly sophisticated representations using advanced machine learning techniques.

When modeling time series, it is essential to consider their temporal characteristics. Time-series data modeling provides natural advantages for improving recurrent neural network (RNN)-based methods [3]. However, the inherent limitations of recurrent models prevent them from running simultaneously. As an alternative, models based on convolutional neural networks were proposed. For sequential data modeling, temporal convolutional networks (TCNs) are a typical representative. This is a characteristic of parallel computing that gives it an advantage. To capture long-term dependence, attention mechanisms are used in another type of temporal context modeling. With the use of self-attention, the transformer successfully illustrates sequential representations.

However, the time-series anomaly detection task is extremely challenging under long-term settings. First, using long-term time series directly requires the extraction of temporal dependencies, which is difficult due to their entangled temporal patterns. We examine the intricate temporal patterns using decomposition, a standard method for time series analysis. Second, since the length of sequences for canonical transformers is quadratic, long-term forecasting cannot be done with them because of their computational complexity.

The model cannot identify reliable relationships due to long-term temporal patterns [4]. Due to the entangled temporal patterns, it is difficult to extract the temporal dependencies directly from the long-term time series. Using decomposition, a standard method for time series analysis, we try to sort out the intricate temporal patterns.

Based on the above information, we propose a decomposition auto-transformer network (DATN) for time-series anomaly detection. DATN still follows residual and encoder-decoder structures. In the encoder layer, we decompose the time series into seasonal and trend components. DATN embeds decomposition blocks as the inner operators. DATN is able to progressively separate long-term trend information by embedding our proposed decomposition blocks as the inner operators. DATN introduces an autoattention module to isolate and represent the periodicity of time series. The frequency-domain dominating periodic patterns are extracted using Fast Fourier Transforms (FFTs) and inverse FFTs.

The contributions of our paper are as follows:

(1) To empower the deep anomaly detection model with imminent decomposition capability, we propose a decomposition auto-attention network as a decomposition architecture;
(2) To extract the dominating periodic patterns in the frequency domain, we propose an auto-attention module to discover the period-based dependencies of time series;
(3) Extensive experimental results on multiple public datasets show that the proposed method achieves state-of-the-art performance.

The remainder of the paper is structured as follows. Section 2 reviews related works. Section 3 examines the problem formulation and Transformer. Section 4 illustrates the proposed decomposition auto-transformer network (DATN). Section 5 describes the experimental process and results analysis of the method. Finally, we summarize the paper.

## 2. Related Work

Anomaly detection in time series has gained an ever-increasing interest in academia and industry [5–7]. There are three categories of existing methods: traditional statistical models, supervised learning methods, and unsupervised learning methods. Statistical models [8,9] such as MA, ARIMA, and Holter Winter are usually used to develop algorithms, and usually make strong assumptions regarding the time series being studied. Different types of time series require different algorithms and fine tuning parameters. As a result, they are not inappropriate for detecting anomalies in complex time series, which are typically found in real applications.

Anomaly detection is considered as a classification problem by supervised learning methods [8,10,11]. A classification model is trained using random forest and detection results are outputted by the aggregation functions. Although the supervised learning methods are powerful and show good results, they require labeled training data, which is difficult in time series anomaly detection. Despite the collection of abnormal data, the data may not include all types of anomalies. Due to this limitation, supervised detection methods have limited applications.

Because of the absence of label information, unsupervised learning methods have received a great deal of attention and popularity. We summarize a few recent works employing RNN, graph, GAN, VAE, and transformer-based methods. RNN-based methods have the advantage of processing time series. In Ref. [3], a variational autoencoder (VAE) was combined with the LSTM to enhance temporal feature representation. In Ref. [12], a stochastic RNN method was proposed to identify anomalies by modeling the data distribution through stochastic latent variables. Graph-based methods are able to model the relationships between signals [13]. In Ref. [14], two parallel GAT modules were employed to capture featurewise and temporal relationships respectively. In Ref. [15], a novel attention-based GNN approach was proposed to learn a graph of dependence relationships between signals. Then, they can identify and explain deviations from these relationships. GAN-based methods [16] train a generator and a discriminator to detect anomalies by the output of the discriminator or the reconstructed error. However, GAN-based methods are typically difficult to train and suffer from the mode collapse problem. In Ref. [17], one encoder and two decoders were designed to reconstruct the input series. Xu et al. [18] proposed an unsupervised anomaly detection algorithm, called Donut, which adopts a CNN as the basic unit for the encoder and decoders, and the training and inference times are sharply reduced. Li et al. [19] proposed a robust and unsupervised anomaly detection algorithm for seasonal key performance indices (KPIs), called Bagel, that adopts CVAE as the basis of the model. Transformer-based methods show great power in sequential data owing to the self-attention mechanism [20]. In Ref. [21], a time series representation learning framework was proposed based on the transformer encoder architecture. In Ref. [22], a self-attention mechanism was renovated to anomaly attention. This mechanism was well designed to represent the prior association and series association of each time stamp. We choose Transformer as the basic framework considering the advantages offered by its multiheaded self-attention.

## 3. Problem Formulation

A time series is generally considered as a collection of observations indexed in time order [6]. It can be defined as $X = \{x_0, ..., x_{T-1}\}$, where $T$ is the length of the time series, $x_t$ is an observation at timestamp $t$, and $0 \leq t < T$. Define the number of signals as $m$, $x_t \in \mathbb{R}^m$ and $X \in \mathbb{R}^{T \times m}$.

Given a training series $X \in \mathbb{R}^{T \times m}$ and a test series $\widehat{X} \in \mathbb{R}^{\widehat{T} \times m}$ with length $\widehat{T}$, the task of unsupervised anomaly detection is to predict an output $Y = \{y_0, ..., y_{\widehat{T}-1}\}$ where $y_t \in \{0, 1\}$, $0 \leq t < \widehat{T}$ to denote whether $\widehat{X}_t$ is an anomaly.

Time series can exhibit a variety of patterns. It is helpful to decompose time series into several components, each representing an underlying pattern. Generally, we consider a time series as the superposition of two different components: a seasonal component, a trend component. The seasonal component at timestamp t, defined as $s_t$, represents the periodic features of the series. The trend component at timestamp t, defined as $p_t$, reflects the long-term progression. The remaining component rt contains anything else in the series. We employ an additive decomposition, then the observation $x_t$ can be written as:

$$x_t = s_t + p_t \tag{1}$$

The transformer [23] is an encoder-decoder structure composed of multiple identical blocks. The multi-headed self-attention module (MHSA) and a feed forward network (FFN) are two essential modules in each block. For a time series $X \in \mathbb{R}^{w \times m}$ within look-back window $w$, its embedding is defined as $E \in \mathbb{R}^{w \times d}$, where $d$ are the dimensions of the embedding features. The attention function firstly projects $E$ into query $Q \in \mathbb{R}^{w \times d(q)}$, key $K \in \mathbb{R}^{w \times d(k)}$, and value $V \in \mathbb{R}^{w \times d(v)}$ with different learned linear projections to $d(q)$, $d(k)$ and $d(v)$ dimensions. Then, we compute their scaled dotproduct attention as follows:

$$Attention(Q, K, V) = Softmax(\frac{QK^T}{\sqrt{d(k)}})V \tag{2}$$

Multi-headed self-attention employs a few different sets of learned projections instead of a single attention. The position-wise feed-forward module is a fully connected network defined as:

$$FFN(A) = LN((ReLU(AW_1 + b_1)W_2 + b_2) + A) \tag{3}$$

where $A$ is the output of the previous layer, $W_1$, $W_2$, $b_1$, and $b_2$ are the trainable parameters, and $LN$ is the Layer normalization operator.

## 4. Decompose Auto-Transformer

The proposed Decompose Auto-Transformer network (DATN) framework is shown in Figure 1. It is constructed by series decompose, auto-attention, multi-head attention, feed forward, and linear decoder. Series decompose block is designed to decompose the time series into seasonal component($X_s$) and trend component($X_t$). *'+'represent feature additive fusion.* With $X_s$ and $X_t$ as input, auto-attention module responsible for embedding multi-scale correlation. Next, the features are fed into the following MHSA. Then, seasonal component embedding and trend component embedding are carried out feature fusion. After linear decoder, reconstruction modules are employed.

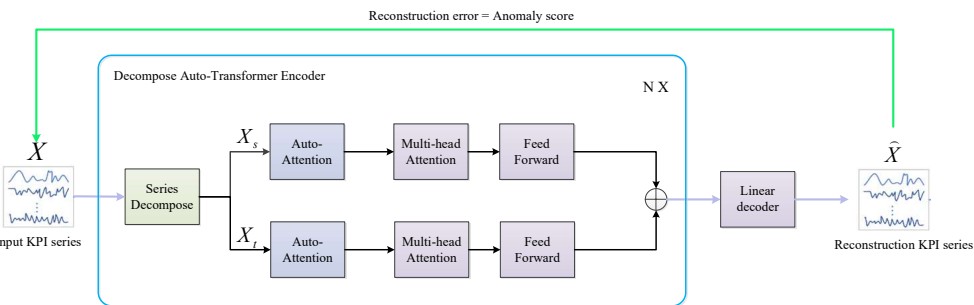

**Figure 1.** Decompose Auto-Transformer network (DATN), framework.

**Series decomposition block.** Time series can exhibit a variety of patterns. It is helpful to decompose time series into several components, each representing an underlying pattern. Generally, we consider of a time series as the superposition of two different components: a seasonal component and a trend component. The seasonal component ($X_s$) represents the periodic features of the series. The trend component $X_t$ reflects the long-term progression. We employ an additive decomposition, then the observation $X$ can be written as:

$$X = X_t + X_s \tag{4}$$

Moving averages are adapted to smooth out the seasonal component and the trend component. For length-$w$ input time series $X \in \mathbb{R}^{w \times m}$, $m$ denotes the time series dimension, and the decomposition formula is as follows:

$$\begin{aligned} X_t &= MA(X) \\ X_s &= X - X_t \end{aligned} \tag{5}$$

where $X_s$, $X_t$ denote the seasonal and trend part, respectively. $MA(\bullet)$ denote the moving average function. We use $X_s, X_t = SeriesDecomp(X)$ to summarize Equations (4).

**Encoder:** As shown in Figure 1, suppose we have $N$ encoder layers. The overall equations for $l$-th encoder layer are summarized as $X^l = Encoder(X^{l-1})$. Details are shown as follows:

$$\begin{aligned} X_s^l, X_t^l &= SeriesDecomp(X^{l-1}) \\ S_s^l &= Auto\_attention(X_s^l) \\ S_t^l &= Auto\_attention(X_t^l) \\ O_s^l &= \text{FFN}(\text{Attention}(S_s^l, S_s^l, S_s^l)) \\ O_t^l &= \text{FFN}(\text{Attention}(S_t^l, S_t^l, S_t^l)) \\ X^l &= O_s^l + O_t^l \end{aligned} \tag{6}$$

where $X^l, l \in \{1, \cdots, N\}$ denotes the output of $l$-th encoder layer. $X_s^l$ and $X_t^l$ denote the seasonal and trend component. $S_s^l$ and $S_t^l$ denote the output of $l$-th $Auto\_attention(\bullet)$. $O_s^l$ and $O_t^l$ denote the output of $Attention(\bullet)$. Attention is a Multi-Headed Self-Attention module. We will give a detailed description of $Auto\_Attention$ in the following paragraph.

**Decode.** Our model reconstructs the time series using the embeddings $X^N$ ($N$ is the number of encoder layers), i.e.,

$$\widehat{X} = Decoder(X^N) \tag{7}$$

where $\widehat{X}$ is the restructure time series, $Decoder(\bullet)$ is a simple linear model.

**Anomaly Criterion.** With the reconstructed time series, we define a score function $s_t$ to flag anomalous behaviors by summing up the prediction errors in time $t$, i.e.,

$$s_t = \sum_{i=1}^{m} \|\widehat{x}_t - x_t\|_2 \tag{8}$$

where $\widehat{x}_t$ is the reconstructed value in time $t$, $x_t$ is input value in time $t$. When the score exceeds a certain threshold, the corresponding data point is identified as an anomaly.

**Auto-Attention.** To extract the period-based feature, we propose an auto-attention module as shown in Figure 2. The input of the $l$-th encoder layer is processed by FFTs (Fast Fourier Transforms). The frequency domain is converted from the time domain. Multi-head attention involves scaled dot-product attention. To discover the dominating periodic patterns, we select top K amplitudes and convert them back to time domain $D_s^l$. Then, multi-head attention is employed to enhance the feature extraction of $D_s^l$. For $l$-layer series $X^l$ is decomposed into seasonal component $X_s^l$ and trend component $X_t^l$. FFT $\mathcal{F}$ and the inverse FFT $\mathcal{F}^{-1}$ are formalized as:

$$
\begin{aligned}
\Gamma_k = \mathcal{F}(x_n) = \sum_{n=0}^{w-1} x_n e^{-\frac{j2\pi}{w}kn} \quad k = 0, ..., w-1 \\
x_n = \mathcal{F}^{-1}(\Gamma_k) = \frac{1}{w} \sum_{n=0}^{w-1} \Gamma_k e^{\frac{j2\pi}{w}kn} \quad k = 0, ..., w-1
\end{aligned}
\tag{9}
$$

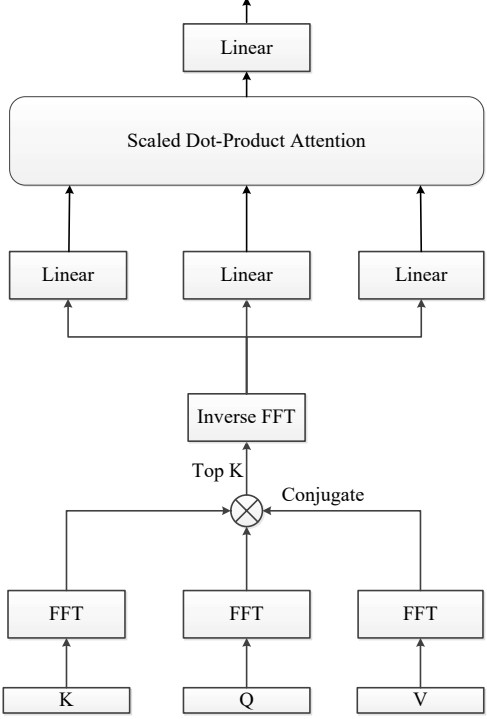

**Figure 2.** The proposed auto-attention module.

For the seasonal component, auto_attention can be expressed as:

$$
\begin{aligned}
D_s^l &= \mathcal{F}^{-1}(Topk(Concat(\mathcal{F}(X_s^l), \mathcal{F}(X_s^l), \mathcal{F}(X_s^l)))) \\
M_s^l &= Attention(D_s^l, D_s^l, D_s^l) \\
Auto\_attention(X_s^l) &= LN(M_s^l)
\end{aligned}
\tag{10}
$$

where $Topk(\bullet)$ obtains arguments of the Topk autocorrelations and let $k = \lfloor c \times \log w \rfloor$, where $c$ is a hyperparameter, $Attention(\bullet)$ is a Multi-Headed self-Attention module, and $LN(\bullet)$ is a linear module. The $Auto\_attention$ of the trend component is the same as the above formula.

## 5. Experiments

### 5.1. Datasets and Model Details

The purpose of this study is to verify the effectiveness of the proposed method, we choose five public datasets:

(1) SMD (Server Machine Dataset) [12] collected from a large Internet company, where each observation is equally spaced by 1 min;
(2) SMAP (Soil Moisture Active Passive satellite) [24] has 562,798 time points, of which the training set size is 135,182 and the testing set size is 427,616;
(3) MSL (Mars Science Laboratory rover) [24] has 132,046 time points, of which the training set size is 58,317 and the testing set size is 73,729;
(4) SWaT (Secure Water Treatment) [25] collected from a real-world water treatment plant with 51 dimensions;
(5) PSM (Pooled Server Metrics) [26] collected from multiple application server nodes at eBay with 26 dimensions.

Dataset statistics are shown in Table 1.

Experimental environment: The operating system is Ubuntu 18.04 LTS, the GPU is a NVIDIA GEFORCE RTX 2080, and the implemented is program is PyTorch 3.6. All experiments are repeated three times.

**Table 1.** Dataset statistics.

| Dataset | Train | Test | Dimensions | Anomalies (%) |
|---------|-------|------|------------|---------------|
| SMD | 708,405 | 708,420 | 38 | 4.16 |
| SMAP | 135,183 | 427,617 | 25 | 13.13 |
| MSL | 58,317 | 73,729 | 55 | 10.72 |
| SWaT | 475,200 | 449,919 | 51 | 12.14 |
| PSM | 132,481 | 87,841 | 25 | 27.76 |

Data normalization and cleaning are first performed on all datasets. We follow the threshold selection method proposed in Ref. [22]. Anomaly scores on the validation subset are clustered to find the threshold. We also employ the point-adjust strategy, which assumes that anomalies are correctly detected in a segment if any anomaly in this segment is detected correctly [12]. Hyperparameter c (in Equation (10)) is set as 4. We use a fixed look-back series $w = 192$ for training and testing. We stack 4 encoder layers in the model. The Adam optimizer is employed with an initial learning rate of $10^{-4}$.

### 5.2. Baselines

We compare our model with seven state-of-the-art baselines:

(1) LSTM-VAE [3]: The model utilizes both VAE and LSTM for anomaly detection;
(2) OmniAnomaly [12]: The model is a stochastic recurrent neural network model that glues Gated Recurrent Unit (GRU) and VAE;
(3) MTAD-GAT [14]: The model considers each univariate time-series as an individual feature and includes two graph attention layers in parallel;
(4) USAD [17]: The model use adversarial training and its architecture allows it to isolate anomalies;
(5) TransFram [21]: The model uses the transformer as the base architecture;
(6) AnomalyTran [22]: The model combines the transformer and the Gaussian prior association makes it easier to distinguish rare anomalies;
(7) TranAD [27]: The model uses an adversarial training program to amplify the reconstruction error, because the simple transformer based networks often miss small abnormal deviations.

These comparison baseline methods are the most representative at present.

*5.3. Performance Comparison*

Precision (P), Recall (R), and F1 score are employed to evaluate the detection performance of all methods. To comprehensively evaluate the five datasets mentioned above, we reproduce part of the results for the baselines. The quantitative evaluation is reported in Table 2, where the best results of the F1 score are in red and the second-best results are blue. From Table 2, we can make the following observations. We achieve the best F1 scores on the SMAP, MSL, SWaT, and PSM datasets and achieve the second-best F1 score on the SMD dataset. Our method achieves high accuracy, which indicates that the false-positive rate is low. Our method performs well in terms of accuracy and recall, with F1 scoring higher than most other methods. Both methods bring significant improvements and new directions. In contrast, we decompose the time series into seasonal components and trend components. Long-term robustness can be retained by auto-attention. We argue that it can improve temporal pattern representation. This hypothesis is confirmed by the results.

**Table 2.** Performance comparison of the baseline methods (as %, red: best, blue: second best).

| Method | SMD | | | SMAP | | | MSL | | | SWaT | | | PSM | | |
|---|---|---|---|---|---|---|---|---|---|---|---|---|---|---|---|
| | P | R | F1 | P | R | F1 | P | R | F1 | P | R | F1 | P | R | F1 |
| LSTM-VAE [3] | 77.63 | 89.12 | 82.98 | 88.65 | 72.18 | 79.57 | 92.32 | 78.61 | 84.92 | 81.78 | 78.55 | 80.13 | 90.16 | 74.48 | 81.57 |
| Omni Anomaly [12] | 83.34 | 94.49 | 88.57 | 74.16 | 97.76 | 84.34 | 88.67 | 91.17 | 89.89 | 86.33 | 76.94 | 81.36 | 91.61 | 71.36 | 80.23 |
| MTAD-GAT [14] | 88.28 | 84.92 | 86.57 | 89.06 | 91.23 | 90.13 | 87.54 | 94.40 | 90.84 | 92.46 | 75.12 | 82.89 | 95.28 | 75.65 | 84.34 |
| USAD [17] | 93.14 | 96.17 | 93.82 | 76.97 | 98.31 | 81.86 | 88.10 | 97.86 | 91.09 | 98.70 | 74.02 | 84.60 | 74.42 | 99.01 | 84.97 |
| TransFram [21] | 91.60 | 86.44 | 88.94 | 85.36 | 87.48 | 86.41 | 97.21 | 90.33 | 93.64 | 92.47 | 75.88 | 83.36 | 88.14 | 86.99 | 87.56 |
| Anomaly Tran [22] | 88.42 | 96.90 | 92.47 | 92.16 | 89.79 | 88.96 | 95.83 | 92.66 | 94.22 | 86.38 | 92.14 | 88.17 | 94.75 | 88.59 | 90.56 |
| TranAD [27] | 92.62 | 99.74 | 96.05 | 80.43 | 99.99 | 89.15 | 90.38 | 99.99 | 94.94 | 97.60 | 69.97 | 81.51 | 88.15 | 84.91 | 89.72 |
| DATN | 93.50 | 94.37 | 93.92 | 96.25 | 86.45 | 91.15 | 96.75 | 93.26 | 94.97 | 95.38 | 85.32 | 90.12 | 97.39 | 87.49 | 92.25 |

To evaluate the efficiency of the proposed method, we provide a comparison of the training time (per epoch) shown in Figure 3. Compared with RNN- and graph-based methods, we need less time per epoch training, as depicted in Figure 3. RNN-based models are weak in parallel computing. The graph-based model is time-consuming due to its complex matrix operations. The graph-based method consumes more time when there are more signals.

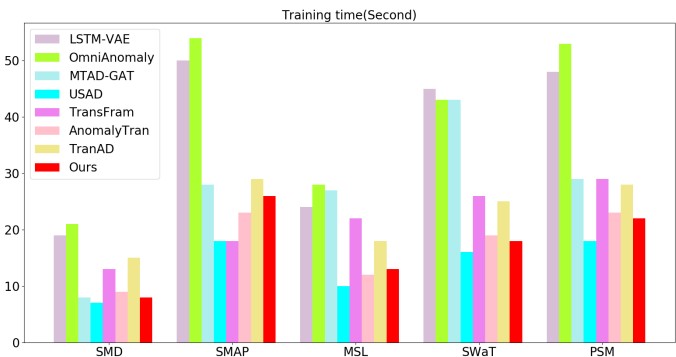

**Figure 3.** Efficiency comparison of the baseline methods.

*5.4. Ablation Study*

We analyze the effectiveness of the main components proposed in the work, including series decompose and auto-attention, as shown in Table 3. For auto-attention, × indicates that the original transformer framework is used. For series decompose, × means that the proposed series decompose is replaced by a simple linear layer. From Table 3, we can make the following observations. Using the proposed series decompose, the average F1 rises from 84.20% to 91.41%. Replacing multi-head attention by our auto-attention, the average F1 rises from 91.41% to 92.45%. That is to say, series decompose and auto-attention can obviously improve the performance by achieving 8.25% (92.45% → 84.20%) and 3.88% (88.08% → 84.20%) absolute average F1 promotions, respectively. In addition, our method

yields a 13.52% (84.20% → 92.45%) absolute average F1 promotion compared with the original Transformer framework.

**Table 3.** Ablation study of series decompose and auto-attention modules.

| Components | | F1 Measures (as %) | | | | | |
|---|---|---|---|---|---|---|---|
| Series Decompose | Auto-Attention | SMD | SMAP | MSL | SWaT | PSM | Avg |
| × | × | 86.21 | 85.23 | 83.13 | 82.21 | 84.22 | 84.20 |
| × | ✓ | 89.23 | 88.14 | 87.19 | 86.36 | 89.43 | 88.08 |
| ✓ | × | 92.38 | 90.38 | 92.75 | 89.17 | 92.15 | 91.41 |
| ✓ | ✓ | 93.95 | 91.13 | 94.99 | 90.05 | 92.15 | 92.45 |

*5.5. Case Study*

We leverage a piece of the SMD dataset for the case study, shown in Figure 4. There are four sub-figures. Figure 4a describes a selected signal with the reconstruction results. The signal is selected due to its greater contribution to anomaly scores. The signal values far from the reconstruction results will generate large anomaly scores (see Figure 4b). Then, with the anomaly scores and threshold, we can easily find the anomalies depicted in the Figure 4c. We produce results essentially in agreement with the anomaly labels shown in the Figure 4d. The differences can be partially avoided by the point-adjust strategy. Generally, the proposed method is effective.

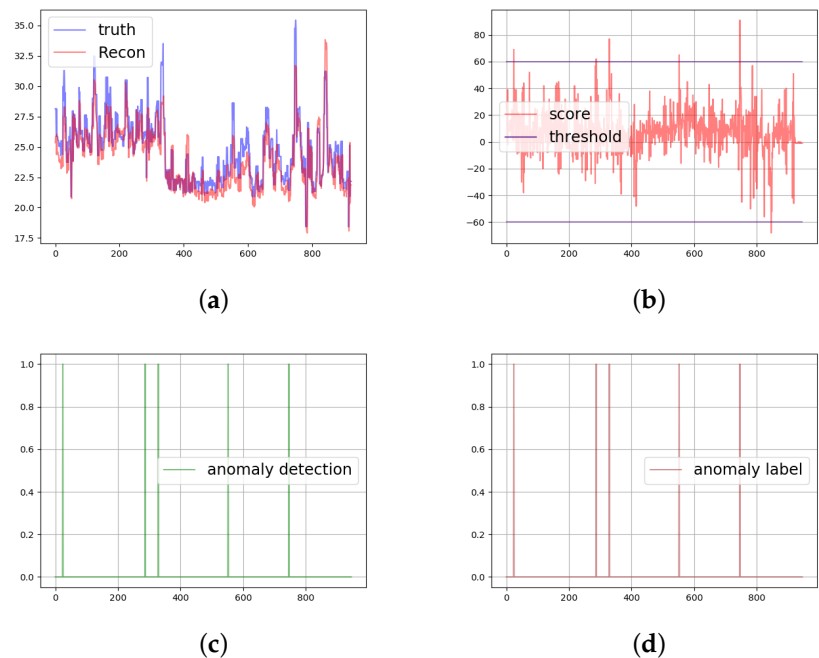

(a)  (b)  (c)  (d)

**Figure 4.** Visualization analysis of anomaly detection. (**a**) The reconstruction results. (**b**) Anomaly scores. (**c**) Anomaly detection results. (**d**) Anomaly labels.

## 6. Conclusions

In this study, we presented a decomposition auto-attention network for time-series anomaly detection. Different from previous methods, we combined the time-series decomposition architecture to learn the intricate temporal patterns in a divide-and-conquer way. Auto-attention based on FFT was carefully designed to isolate and represent the periodicity of time series. We proved the effectiveness of our detection algorithm on five real datasets. These experiments demonstrated the efficacy and robustness of DATN and the key module auto-attention, promising general use for time series reconstruction without the need for model customization and greedy hyperparameter tuning. The only limitation is the requirement of a certain amount of training data, as in typical neural networks.

In future work, we plan to expand our DATN to other intelligent tasks for IoT systems including forecasting and smart control. We also plan to make use of the feature extraction ability of auto-attention and explore an alternative usage of auto-attention from the perspective of representation learning for time series.

**Author Contributions:** Conceptualization, Y.C.; Data curation, C.F.; Funding acquisition, Y.T.; Methodology, B.W.; Resources, Z.Y. All authors have read and agreed to the published version of the manuscript.

**Funding:** This research was funded by Opening Foundation of Key Laboratory of Computer Network and Information Integration (Southeast University), Ministry of Education grant number K93-9-2021-05.

**Conflicts of Interest:** The authors declare no conflict of interest.

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
