# Peer review of "Decompose Auto-Transformer Time Series Anomaly Detection for Network Management†"

_electronics, doi:10.3390/electronics12020354_

Round 1

Reviewer 1 Report

General Comments:

1. This manuscript is NOT well prepared and in poor English writing. In addition, there are many grammatical and typo errors in the manuscript. 

2. Some references are NOT adequate. For example, the reference on [1] for "time series anomaly detection" is NOT adequate. Please consider to add the following references:

Shaukat, K., Alam, T. M., Luo, S., Shabbir, S., Hameed, I. A., Li, J., ... & Javed, U. (2021, April). A review of time-series anomaly detection techniques: A step to future perspectives. In Future of Information and Communication Conference (pp. 865-877). Springer, Cham.

Choi, K., Yi, J., Park, C., & Yoon, S. (2021). Deep learning for anomaly detection in time-series data: review, analysis, and guidelines. IEEE Access.

Sgueglia, A., Di Sorbo, A., Visaggio, C. A., & Canfora, G. (2022). A systematic literature review of IoT time series anomaly detection solutions. Future Generation Computer Systems.

Shah, S. H. A., Akbar, M. J., & Raza, U. A. (2021). A Review on Anomaly Detection in Time Series. International Journal10(3).

3. The Sections of 1. Introduction and 6. Conclusion are too short, simple and weak to describe the motivation and contribution of this research. Please improve.

4. Why does this paper deal with "resource management" problems? The terms of "resource", "management", and "resource management" do NOT appear in the whole context (including the Abstract). How does the developed DATN approach tackle the resource management or resource allocation problems?

Questions/Suggestions

Line 3

The representation, reconstruction of time series have ...

=> The representation and reconstruction of time series have ...

Lines 4 and 6

The authors wrote "However, time series are ..." in Line 4 and "Time series is ..." in Line 6. 

Question: Is "time series" single or plural? Please check.

Line 9

... a auto-transformer ...

=> ... an auto-transformer ...

Line 10

... on series seasonal and trend.

=> ... on series seasonal and trend components.

Line 11

... by the simple linear decoder, ...

=> ... by a simple linear decoder, ...

Lines 78~79

Dount [12] adopted ...

=> Xu et al. [12] proposed unsupervised anomaly detection algorithm, Donut, that adopts ...

Line 80

Bagel[13] adopted ..

=> Li et al. [13] proposed a robust and unsupervised anomaly detection algorithm for seasonal key performance indices (KPIs), Bagel, that adopts ...

Eq. (1), Page 3

Please define or provide references to the functions of Soft and max in Eq. (1).

Line 159

The authors wrote "Hyper-parameter c (in Equation 9) set 4."

Question: What does this sentence describe?

Line 160

Adam optimizer 

=> The Adam optimizer

5th row, 1st Column, Table 1, Page 6

sWaT 

=> SWaT

Lines 183-185

The authors wrote "..., as depicted in Fig. 3. RNN-based models are weak in parallel computing. Graph-based model is time-consuming due to the complex matrix operations. Graph-based method consume more time when there are more signals."

Question: Why? How does Fig. 3 support these observations and claims?

Line 184

Graph-based model is ...

=> Graph-based models are ... 

Line 185

Graph-based method

=> Graph-based methods consume ...

Author Response

The co-anthors and I would like to thank you for the time and effort spent in reviewing the manuscript. Each comment will be directly addressed regrading the modified manuscript with changes in red.

Reviewer 2 Report

The related work survey is not complete enough. The authors should provide the related studies that attack the same problem and make a comparison with the results reported in the related studies. The authors simply implemented their own versions for the "competitors", which may not be optimal at all. The results are misleading.

Author Response

(The authors gave the same response as above.)

Reviewer 3 Report

My questions an suggestions:

1. In the section Related works there is not motivation why this research is needed at all. There are many other methods to find anomalies in time series (7 methods in the Table 2) and most of them are.efficient enough. Why do we need one more method? The reason is not clearly described  in the article.

2. Top K is probably means some kind of filter? Which kind? What is about phase? Selection of several samples in frequency domain will lead to strange and probably inadequate result in time domain.

3. FFT works on series of samples in time domain. What is the length of this series?

4. There is no Discussion section. Also there's no disadvantages if the proposed methods. Is it a new silver bullet?

Author Response

(The authors gave the same response as above.)

Round 2

Reviewer 1 Report

General Comments:

1. The English writing in this revised manuscript is still NOT qualified for publication. There are many grammatical errors. The readability of this article is poor. Most descriptions in the manuscript are either lack of words or difficult to understand. 

2. Why is the proposed approach good (or as claimed by the authors, the state-of-the-art) for "resource management"? Although the authors added sentences (Lines 17~22) for "resource management", why is "efficient anomaly detection technology can avoid resource waste (Lines 21-22)" important for resource management? Why do the contributions of this research NOT include any of "resource management", as claimed in its paper title?  Please add more descriptions on the contributions of this research to resource management problems/issues. Please also consider to add more references for anomaly detection in resource management.

3. Please consider to reorganize the entire section of Section 3 PRELIMINARIES (including Subsections 3.1 Problem Formulation and 3.2 Transformer), combine these two subsections and replace it with Section 3 PROBLEM FORMULATION, a more clear presentation of problem descriptions, the approach of decomposition of time series, encoding/decoding mechanisms, and so on.

4. In Line 199, the authors described "We stack 4 encoder layers in the model." Why? There are NO "stacked" encoder architecture or design presented in the manuscript.

5. Please consider to reorganize the 5. EXPERIMENTS section and make it more clear in the descriptions of Experiment Design, Datasets for Baseline Studies, Numerical Results, Ablation and Case Studies, and more. Subsections 5.2, 5.4 and 5.5 of this revised manuscript are too short and lack of words to support the claimed contributions.

Questions/Suggestions

Line 10

... based on series seasonal and trend components.

Question: ??? What does this mean?

Line 177

To discovers the dominating ...

=> To discover the dominating ...

Lines 202~205

The authors wrote "We compare our model with 7 state-of-the-art baselines, ..."

Questions: Why are these 7 methods selected for comparisons? Please add more descriptions about the needs, reasons and motivations to adopt these seven benchmarking methods. 

Line 207

Precision (P), recall (R) and F1 score are ...

=> Precision (P), Recall (R) and F1 scores are ...

Figure 4, Page 8

Suggestion: Please separate the figure into three sub-figures--4(a), 4(b) and 4(c), and add captions to each subfigure.

Lines 230-231

Please move the descriptions of " ? denotes employing series decompose or Auto-Attention module. × denotes not using series decompose or Auto-Attention module." to the captions of Table 3.

Line 232

With complete DATN model strategy, 

=> With the complete DATN model strategy, 

Line 233

5%(88.08 → 92.45) and 1% (91.41 → 92.45) 

=> 5%(88.08% → 92.45%) and 1% (91.41% → 92.45%) 

Lines 237~238

The authors wrote "The first one (from top to bottom) describes the anomaly scores and threshold, we can easily find the anomalies depicted in the second sub-figure." 

Question: ??? Grammatical errors in this long sentence.

Author Response

    1. The English writing in this revised manuscript is still NOT qualified for publication. There are many grammatical errors. The readability of this article is poor. Most descriptions in the manuscript are either lack of words or difficult to understand. 

    Response:We appreciate it very much for this good suggestion. In order to improve the language of the manuscript, we adopted AJE to polish the entire manuscript.

    1. Why is the proposed approach good (or as claimed by the authors, the state-of-the-art) for "resource management"? Although the authors added sentences (Lines 17~22) for "resource management", why is "efficient anomaly detection technology can avoidresource waste (Lines 21-22)" important for resource management? Why do the contributions of this research NOT include any of "resource management", as claimed in its paper title?  Please add more descriptions on the contributions of this research to resource management problems/issues. Please also consider to add more references for anomaly detection in resource management.

    Response:We appreciate it very much for this good suggestion. We change the goal of the paper to network intelligent management, and anomaly detection is the key technology of network intelligent management.

    1. Please consider to reorganize the entire section of Section 3 PRELIMINARIES (including Subsections 3.1 Problem Formulation and 3.2 Transformer), combine these two subsections and replace it with Section 3 PROBLEM FORMULATION, a more clear presentation of problem descriptions, the approach of decomposition of time series, encoding/decoding mechanisms, and so on.

    Response:We appreciate it very much for this good suggestion. In the revised version, we have made changes according to your comments.

    1. In Line 199, the authors described "We stack 4 encoder layers in the model." Why? There are NO "stacked" encoder architecture or design presented in the manuscript.

    Response:We appreciate it very much for this good suggestion. N X in Figure 1 indicate stacked.

    1. Please consider to reorganize the 5. EXPERIMENTS section and make it more clear in the descriptions of Experiment Design, Datasets for Baseline Studies, Numerical Results, Ablation and Case Studies, and more. Subsections 5.2, 5.4 and 5.5 of this revised manuscript are too short and lack of words to support the claimed contributions.

    Response:We appreciate it very much for this good suggestion. In the revised version, we have made a more detailed description in the experiment sections. 

Reviewer 2 Report

My comments have been addressed.

Author Response

none

Reviewer 3 Report

All of my questions were answered good enough and the new revision of the article obtained more scientific value.

Author Response

none

Round 3

Reviewer 1 Report

Suggestions

1. Please add some references on "Time Series Anomaly Detection" as it is "a widely used technology" as stated in the first sentence, the first paragraph, the first section of this article.

2. Please prepare, review and review again the manuscript before its submission. For example, in Lines 275, 277~279, the terms of "Fig. ??" should not appear in its latest revision.

Author Response

  1. Please add some references on "Time Series Anomaly Detection" as it is "a widely used technology" as stated in the first sentence, the first paragraph, the first section of this article.

Response:We appreciate it very much for this good suggestion. 

In the revised version,we add some references and describe them.

2. Please prepare, review and review again the manuscript before its submission. For example, in Lines 275, 277~279, the terms of "Fig. ??" should not appear in its latest revision.

Response:We appreciate it very much for this good suggestion.  In the revised version,we checked the manuscript repeatedly and revised it according to the comments.